# Health workforce for oral health inequity: Opportunity for action

**Jennifer E. Gallagher** [1]*, **Grazielle C. Mattos Savage**[1], **Sarah C. Crummey**[1],
**Wael Sabbah**[1], **Yuka Makino**[2], **Benoit Varenne**[3]

1 Dental Public Health, Centre for Host Microbiome Interactions, King's College London, Faculty of Dentistry, Oral & Craniofacial Sciences, Denmark Hill Campus, London, United Kingdom, 2 Noncommunicable Diseases Management Team, WHO Regional Office for Africa, Cité Djoué, Brazzaville, Congo, 3 WHO Oral Health Programme, Noncommunicable Diseases Department | Division of Universal Health Coverage & Communicable and Noncommunicable Diseases, World Health Organization, Geneva, Switzerland

* jenny.gallagher@kcl.ac.uk

## Abstract

Oral health is high on the global agenda following the adoption of the 2022 global strategy on oral health at the 75th World Health Assembly. Given the global burden of oral disease, workforce development to achieve universal health coverage [UHC] is crucial to respond to population needs within the non-communicable disease agenda. The aim of this paper is to present an overview of the oral health workforce [OHWF] globally in relation to key contextual factors. Data from the National Health Workforce Accounts and a survey of World Health Organization [WHO] member states were integrated for analysis, together with country-level data on population and income status. Data are presented using the WHO categorisation of global regions and income status categories established by the World Bank. Workforce densities for key OHWF categories were examined. Multiple regression was used to model workforce density and contextual influences. Challenges and possible solutions were examined by country income status. There are approximately 3.30 dentists per 10,000 population globally, and a combined OHWF [dentists, dental assistants/therapists and dental prosthetic technicians] of 5.31 per 10,000. Marked regional inequalities are evident, most notably between WHO European and African regions; yet both make greater use of skill mix than other regions. When adjusted by region, 'country income status' and 'population urbanization' are strong predictors of the workforce density of dentists and even more so for the combined OHWF. Maldistribution of the workforce [urban/rural] was considered a particular workforce challenge globally and especially for lower-income countries. Strengthening oral health policy was considered most important for the future. The global distribution of dentists, and the OHWF generally, is inequitable, with variable and limited use of skill mix. Creative workforce development is required to achieve the global oral health agenda and work towards equity using innovative models of care, supported by effective governance and integrated policies.

**Data Availability Statement:** Data availability, by country and professional category, is reported in S1 Table and the dataset is available at S2 Table.

**Funding:** The author(s) received no specific funding for this work.

**Competing interests:** The authors have declared that no competing interests exist.

## Introduction

A 'landmark global strategy on oral health' [1] adopted at the 75[th] World Health Assembly [WHA] in May 2022 as part of the noncommunicable disease [NCD] strategy [2], follows the commitment of World Health Organization [WHO] member states to bring oral health back to the top of the global health agenda [3]. It builds on previous declarations on NCDs [4], and Universal Health Coverage [UHC] [5], in support of global Sustainable Development Goals [SDGs] [6]. Given the high burden of untreated oral diseases and conditions worldwide, affecting almost 3.5 billion people [7, 8], urgent action is required. Oral diseases, most notably dental caries and periodontal diseases, are associated with other major NCDs, by sharing common risk factors, and responsible for significant social, economic and health burdens [7, 9, 10].

The global strategy which includes the goal of achieving "UHC for oral health for all individuals and communities by 2030, enabling them to enjoy the highest attainable state of oral health and contributing to healthy and productive lives" [1], requires human resources to facilitate adequate health care delivery [2, 11].

The importance of policies in addressing oral health and oral health workforce disparities between, and within, countries is well recognised; together with the need to manage, develop, and retain them [2, 3, 5, 12]. The COVID-19 pandemic has highlighted that whilst all health systems have been stretched, under-staffed health systems struggle to respond effectively to the broad range of new and old population health needs [13]. Even in high income countries, there is evidence that dental services, particularly in the public sector, are stretched to capacity [14].

In support of building the global workforce for 2030, there has been progressive action to strengthen information systems [15], with increasing emphasis on the oral health workforce [16, 17], to inform evidence-based national and international policies [2, 3, 11, 12, 16, 17]. The National Health Workforce Accounts [NHWA] play an important role in building national capacity to strengthen and facilitate data sharing and use [18], including standardization of health workforce information systems for interoperability, that is the ability to exchange data within broader subnational or national health information systems, as well as within international information systems. They increasingly serve as a guide and support-tool informing national evidence based workforce policy decisions [15]. Furthermore, health personnel planning and development, based on the WHO Global competency and Outcomes framework [19], are increasingly advocated.

Within this paper, the term *oral health workforce* [OHWF] will be used to cover four occupational categories in line with the International Standard Classification of Occupations [ISCO-08] [20]: dentists [2261], dental assistants/therapists [3251], dental prosthetic technicians [3214], and dental aids [5329]; the global definitions of which are outlined in S1 Text. Dentists require the longest education and training of all oral health providers [20, 21], and are the traditional lynch-pin of the dental team. Dental assistants/therapists are considered mid-level providers [22–24], and this group includes dental hygienists and dental nurses who may provide clinical care [20]. Some dental prosthetic technicians may also provide clinical care [20]. Whilst the occupation of 'dental aides' [sometimes called dental nurses or dental surgery assistants [involved in organisation of the dental surgery, and infection prevention and control, together with preparing mixing and handling dental materials] is considered vital amongst oral health personnel [20], they are largely unregistered. Data on 'dental aides' are therefore limited or non-existent in much of the world, whilst in some countries such as the United Kingdom and Brazil, their role has been professionalised as a 'dental nurse' and 'oral health assistant' respectively, with all required to be registered, or in training, as they are

involved in patient care [25, 26]. Furthermore, there are variations in categorisation and scope of practice adding to complexity of OHWF terminology and professional 'scope'.

As member states take account of the new global strategy for oral health [1], there is an urgent need to consider OHWF patterns and trends, taking account of contextual factors, in support of planning [17] for UHC [5].

## Aim

To examine the profile and trends in the global OHWF by country, WHO region, income status and level of urbanization, together with dental education provision, workforce challenges and possible solutions.

## Materials and methods

Data on 192 of the 194 WHO member states were available from two sources. First, trend data extracted from the National Health Workforce Accounts [NHWA] [18] for the years 2000–2019. Second, the findings of a global oral health survey of member states conducted in 2019 [June-December], as a collaborative project between the WHO and King's College London. The survey was distributed to countries through WHO regional offices in English, French, Spanish and Russian. The scope of the survey was aligned with ISCO workforce categories [20]. Using a mix of open and closed questions the questionnaire instrument, available in S2 Text, explored the capacity, capability, education, and training of the OHWF, and perceived challenges/solutions. The questionnaire was developed by WHO and King's College London teams against standard criteria outlined in the NHWA [18], together with the wider literature, and tested with WHO oral health and workforce experts for face validity. This survey was exempt from Ethics Research Committee review because the scope of our research was clearly public health activities, involving public health [workforce] surveillance, and not human participant research, based on WHO guidance.

Extracted NHWA data covered dentists, dental assistants/therapists, and dental prosthetic technicians; absolute numbers, and density per 10,000 population, the past two decades. From the global survey, absolute numbers of the same categories of personnel were used to calculate density per 10,000 population using member states population data from 2019 [27]:

$$\frac{Number\ of\ personnel \times 10,000}{Country\ population}$$

For further details please see S1 and S2 Tables.

A combined study database was thus created through triangulation of the available information outlined above. Latest data reports of member states ranged between 2002 and 2019. NHWA data were used as the source of choice where data were similar. In the case of discrepancies between data sources for 2019, additional evidence was sought by the WHO from members states, where required, to ensure a robust contemporary dataset and a final check conducted March 2021 for the latest 2019 data recorded in the NHWA.

Respondents to the global survey comprised 112 WHO member states [57.73%], the WHO African region [AFR] having the highest response [85.11%] and the WHO American region [AMR] and the WHO Eastern Mediterranean region [EMR] joint lowest [28.57%] responses. Whereas dentists are present in almost all countries, many do not provide evidence of having other categories of personnel. Thus, across the 194 WHO member states, data availability was high for dentists [98.97%; n = 192], and lower for dental assistants/therapists [67.53%; n = 131], and dental prosthetic technicians [64.95%; n = 126]. Regionally, data coverage for

dental assistants/therapists ranged from 47.62% in EMR to 87.23% in AFR, and for dental prosthetic technicians from 45.71% in AMR to 100% in the WHO South-East Asia [SEA].

Using this more comprehensive database, descriptive analysis of the OHWF examined data coverage and workforce densities per 10,000 population, together with education levels, perceived workforce challenges and possible solutions. Information on levels of urbanization and income status [high, upper-middle, lower-middle, and low] were obtained for member states from the United Nations Human Development Program [28], and The World Bank [29], respectively, and combined with workforce data for analysis.

First, descriptive analysis of OHWF density was conducted by WHO region [Africa = AFR, Americas = AMR, Eastern Mediterranean = EMR, Europe = EUR, South-East Asia = SEA, and Western Pacific = WRP]; and second, by income status [high, upper-middle, lower-middle, low-income]. Third, trends over time were examined by region and for the world's most populous countries. Forth, multiple regression analysis tested the association between the independent variables: density of dentists and density of the combined OHWF, and dependent variables, 'percentage urban population' and 'income status', adjusted by region. Fifth, evidence on education establishments by region and income status were examined. Sixth survey data relating to challenges and solutions were analysed by income status and trends assessed using Analysis of Variance [ANOVA]. Qualitative data, obtained through responses to open questions within the global survey, were thematically analysed [30, 31], to provide additional insights to perceived challenges and possible solutions.

Data management was performed using Microsoft Excel 365 license 10030000A0BC931E and statistical analysis was performed using the Statistical software for data science (Stata)' 17 serial number 501709308936 and the Statistical package for the social sciences (SPSS) Version 27.

## Results

### Oral health workforce [OHWF] globally

There were 2.49M dentists recorded across 192 countries in 2019; 1.24M dental assistants/therapists across 131 countries; and 0.28M dental prosthetic technicians across 126 countries, giving a total of 4M oral health workers globally [Table 1].

On average, there are approximately 3.30 dentists per 10,000 population globally, ranging from only 0.44 in AFR to 5.91 in EUR. For dental assistants/therapists and dental prosthetic technicians, the average worldwide ratios are 2.63 per 10,000 and 0.59 per 10,000, respectively, making a combined OHWF of 5.31 per 10,000 overall. Considering the use of skill mix, as opposed to dentists, the global ratio of dentists to dental assistants/therapists is 1: 0.80 and dentists to dental prosthetic technicians lower at 1: 0.18.

There is over a 10-fold difference in the density of dentists between high- [7.05; n = 60], and low-income [0.55; n = 28] status countries, as shown in Table 2. Interestingly, this difference between high- and low-income countries is much more pronounced for the mid-level workforce. For dental assistants/therapists there is a 176 times greater proportion of these professionals reported in high-income than low-income countries, and a 37-fold difference for dental prosthetic technicians.

San Marino [17.76] in Europe, with a population of 33,864 inhabitants [32], has the highest density of dentists per 10,000 population in the world followed by Cuba [16.70], and Argentina [15.35]. All five states with the lowest densities of dentists in the world are in AFR: namely South Sudan [0.003], Burundi [0.004], Chad [0.007], Mali [0.007], and Togo [0.008].

**Table 1. Density—Global OHWF per 10,000 population, by category and region, 2019[1−3].**

| Region | Population | Dentists Densities | Number | Number of countries reporting/ total | Dental Assistants/ Therapists Densities | Number | Number of countries reporting/ total | Dental Prosthetics Technicians Densities | Number | Number of countries reporting/ total |
|---|---|---|---|---|---|---|---|---|---|---|
| | | | | | Capacity and coverage | | | | | |
| EUR | 921M | 5.91 | 549,285 | 53/53 | 10.35 | 304,053 | 28/53 | 3.97 | 100,949 | 26/53 |
| AMR | 1,010M | 5.67 | 572,346 | 35/35 | 7.90 | 659,316 | 22/35 | 1.35 | 65,076 | 16/35 |
| WRP | 1,923M | 4.55 | 830,309 | 26/27 | 5.37 | 210,815 | 21/27 | 2.05 | 73,678 | 20/27 |
| EMR | 712M | 2.50 | 174,491 | 20/21 | 0.27 | 12,608 | 10/21 | 0.24 | 13,775 | 11/21 |
| SEA | 2,002M | 1.64 | 328,998 | 11/11 | 0.13 | 23,977 | 9/11 | 0.06 | 11,533 | 11/11 |
| AFR | 1,092M | 0.44 | 36,222 | 47/47 | 0.39 | 32,783 | 31/47 | 0.10 | 11,986 | 31/47 |
| Global | 7,667M | 3.30 | 2,491,651 | 192/194 | 2.63 | 1,243,552 | 131/194 | 0.59 | 276,997 | 126/194 |

Notes

[1] SOURCE: The workforce data are based on the latest available data in the NHWA data platform as of 31 March 2021, reporting the latest available data for 2019. The oldest data relate to 2002. The Philippines is the only exception where valid data for 2020 have been reported and used.

[2] According to the International Standard Classification of Occupations [ISCO-08] https://www.ilo.org/wcmsp5/groups/public/---dgreports/---dcomm/---publ/documents/publication/wcms_172572.pdf, categories of professionals were defined. Under this dental assistants and therapists, dental hygienists and dental nurses were covered.

[3] Data coverage for all oral health workforce in %—Dentists: 100% for Member States in AFR, AMR, EUR, SEA Regions, and over 95% for Member States in EMR and WRP. Dental assistants/therapists: 87.23% for AFR, 62.86% for AMR, 47.62% for EMR, 52.83% for EUR, 81.81% for SEA, and 77.78% for WRP. Dental prosthetic technicians: 89.36% for AFR, 45.71% for AMR, 52.38% for EMR, 49.06% for EUR, 100% for SEA, and 74.07% for WRP.

For dental assistants/therapists, the countries with the highest densities in the world are San Marino [42.818], Germany [27.08], and South Korea [15.47], followed by the United States [15.05] and Canada [14.28].

For dental prosthetic technicians, the highest densities are Niue [12.39], Germany [8.14], South Korea [6.82], San Marino [5.91] and Luxembourg [5.39].

Over the past decade, the global dentist workforce grew by 95% [Fig 1]. Interestingly, most of the growth occurred in the Western Pacific Region [WPR], which almost quintupled its capacity.

The 15 most populous countries globally [Fig 2], represent around two thirds of the world population, and reveal marked inequalities and changes over time in dentists. Japan has the highest density of dentists [7.95], followed by Brazil [6.40], the United States [6.10], and China [4.46]. In contrast, Bangladesh [0.59], Indonesia [0.58], Nigeria [0.22], and Ethiopia [0.16], consistently have less than one dentist per 10,000 population. There are unexplained abrupt changes in the trend patterns amongst these populous countries. Most notably Brazil, where reported dentist density fell by almost half in 2018 [compared with 2017], whilst in China it increased five-fold in 2017, following a six-year period with no updates.

## Dental education institutions

Dental education is delivered in both the public and private sectors. The global survey identified 930 dental schools worldwide across 100 [51.55%] member states as shown in S3 Table. Most dental schools are in middle income countries [69%] and private [60%]. Whilst EUR and AFR, reported a higher volume of public dental schools, all other regions reported a higher volume of private schools. SEA and AFR provide marked contrasts, with SEA reporting having four times more private than public schools, whilst AFR, reported four times more public dental schools than private. The global survey also identified 53 countries having community

**Table 2. Density—Global OHWF per 10,000 population, by category and country income status, 2019[1−3,5].**

| Income Status[4] | | Capacity and coverage | | | | | | | | |
|---|---|---|---|---|---|---|---|---|---|---|
| | Population number % | Dentists Densities | Number | Number of countries reporting/ total | Dental Assistants/ Therapists Densities | Number | Number of countries reporting/ total | Dental Prosthetics Technicians Densities | Number | Number of countries reporting/ total |
| High | 1,197M | 7.05 | 844,358 | 60/60 | 12.36 | 1,052,849 | 41/60 | 3.70 | 172,901 | 33/60 |
| | 15.6% | | | | | | | | | |
| Upper-middle | 2,895M | 4.04 | 1,168,525 | 54/54 | 1.45 | 149,271 | 33/54 | 0.80 | 82,155 | 34/54 |
| | 37.7% | | | | | | | | | |
| Lower-middle | 2,909M | 1.57 | 442,738 | 48/49 | 0.16 | 38,110 | 36/49 | 0.06 | 16,458 | 34/49 |
| | 37.9% | | | | | | | | | |
| Low | 668M | 0.55 | 36,022 | 28/29 | 0.07 | 3,314 | 20/29 | 0.10 | 5480 | 25/29 |
| | 8.7% | | | | | | | | | |
| Global | 7,667M | 3.30 | 2,491,643 | 190/192 | 2.63 | 1,243,544 | 130/192 | 0.59 | 276,994 | 126/192 |
| | 100% | | | | | | | | | |

**Notes**

[1] SOURCE: The workforce data are based on the latest available data in the NHWA data platform as of 31 March 2021, reporting the latest available data for 2019. The oldest data relate to 2002. The Philippines is the only exception where valid data for 2020 have been reported and used.

[2] According to the International Standard Classification of Occupations [ISCO-08]

https://www.ilo.org/wcmsp5/groups/public/---dgreports/---dcomm/---publ/documents/publication/wcms_172572.pdf, categories of professionals were defined. Under this dental assistants and therapists, dental hygienists and dental nurses were covered.

[3] Two countries from 194 participants had no record for income status: Cook Islands/ Niue–excluded from analysis.

[4] Countries change their income status over time. This document is based on the countries current income status provided by the GHWA.

[5] Data coverage for all oral health workforce in %—Dentists: 100% for Member States

[6] Population by income status [2019 data].

health workers involved in oral health activities: 17 in AFR, 12 in WRP, 10 in EUR, 8 in AMR, 5 in SEA and 1 in EMR. Overall, 17% of them had low-income status, 34% lower-middle, 24.5% upper-middle and 24.5% belong to the high-income status bracket.

## Regression analysis

Regression analyses of the global data revealed significant differences between AFR and all other regions, the latter having significantly higher density of dentists in dentists per 10,000 population [Table 3] and of the combined OHWF [Table 4]. EUR's rate of dentist-density is over six times higher than AFR.

*Levels of urbanization* and *income status* of member states are predictors OHWF density as outlined below. First, the more urbanised a population is, the more likely it is to have dentists; and it is even more likely it is to have an OHWF. For every unit increase in the urban population, there is a 5.67 times higher rate ratio of dentists per 10,000, and a 6.57 times higher rate ratio for the combined OHWF; hence, urbanization is associated with higher OHWF densities and not merely dentists. Second, the income status of a country is a significant predictor of dentist density and oral health professional density. Countries with high-income and upper-middle income status had higher rate ratios of dentists [2.23; 1.86, respectively], compared with low-income status [1.00] [Table 3]. When considering the combined oral health workforce [Table 4], there is a more marked gradient, and greater difference, between low-income and high-income status countries with the latter even more likely to have the entire oral health team together [rate ratio 3.76], than just dentists [rate ratio 2.23].

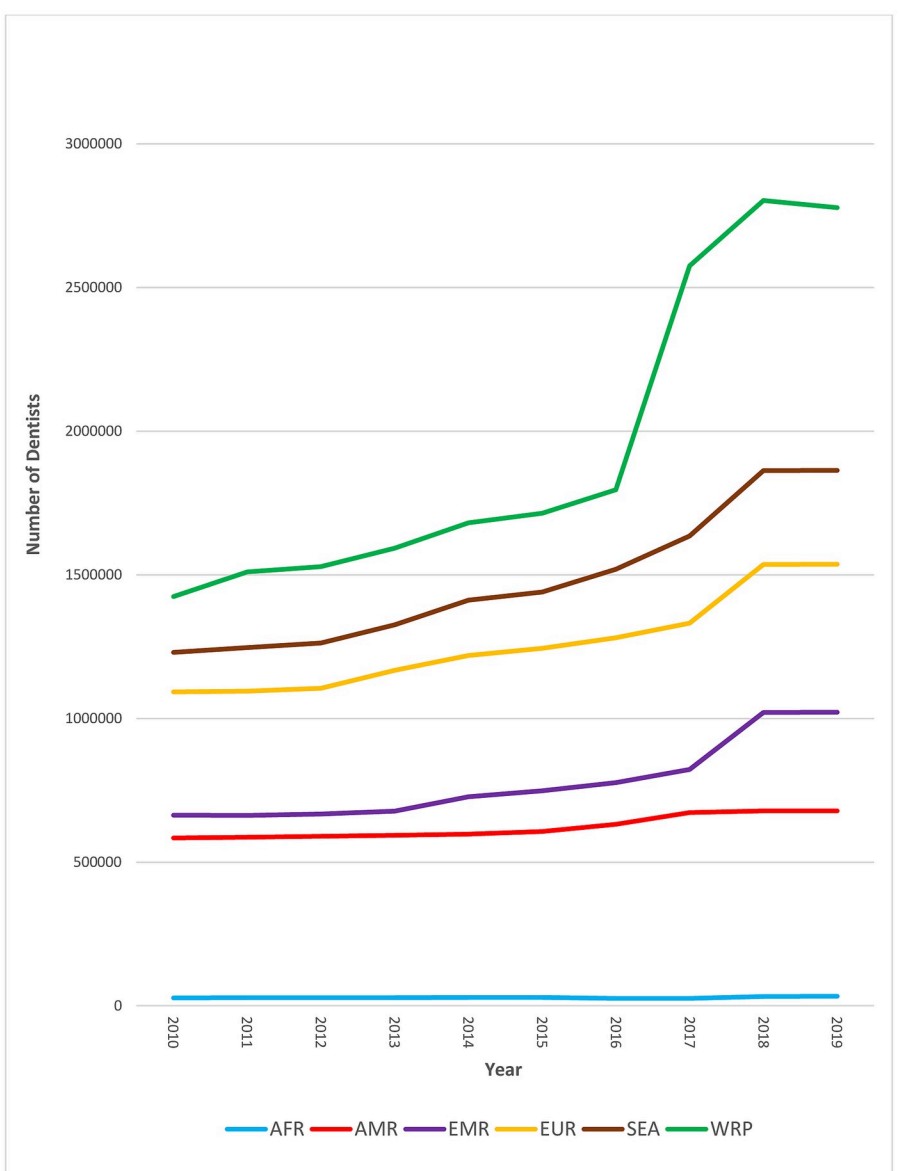

**Fig 1. Trends in global number of dentists by WHO region, 2010–2019.** Source: The workforce numbers are based on the latest available data in the NHWA data platform as of 31 March 2021, apart from the data for 2019, which is combination of the latest available data from the NHWA data platform and WHO/King's College London 2019 survey.

## Perceived workforce challenges

A wide range of challenges was identified by the 95 countries responding to the global survey, led firstly by 'maldistribution of the workforce [urban/rural]'; secondly 'oral health' being 'considered low priority'; and thirdly by 'lack of financial support for OHWF training institutions' [Table 5]. All three showed a clear significant pattern by income status, with lower income countries being more likely to report these issues as challenges than higher income countries. In addition to 'lack of financial support for OHWF training institutions', several further challenges were highly significant by income status: 'lack of continuing professional development opportunities', the 'existence of unregistered providers of dentistry', and 'poor quality dental

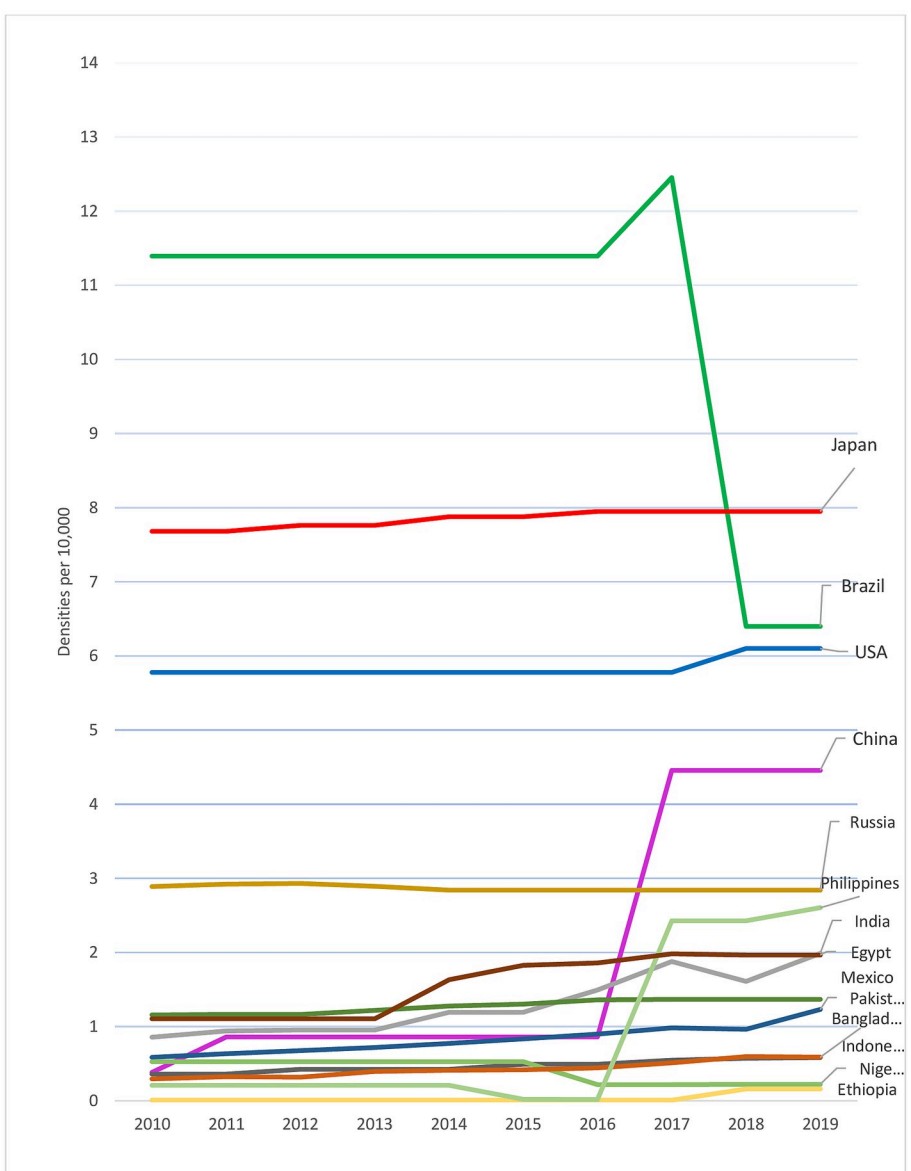

**Fig 2. Trends in density of dentists per 10,000 population, top 15 most populous countries, 2000–2019.** Source: The workforce data are based on the latest available data in the NHWA data platform as of 31 March 2021, apart from the data for 2019, which a combination of the latest available data from the NHWA data platform and WHO/King's College London survey. Countries with the highest densities were named in the plot and highlighted in bold.

care'. Additionally, higher income states were less likely to perceive 'training too many dentists' as a presenting challenge.

Other challenges highlighted by countries [n = 26] in response to open questions, shared common themes including the 'lack of oral disease control programmes' and 'lack of funding' for such programmes and 'inadequate equipment'. There were concerns over having 'no training structure for [professionals] in charge of care', and 'training too few dental assistants' to support the delivery of quality care.

**Table 3. Regression analysis of dentists per 10,000 population by region.**

| Density of dentists[1] | Rate Ratio | Sig. | 95% conf. interval |
|---|---|---|---|
| **WHO Region** | | | |
| **AFR (*reference category*)** | 1 | | |
| **AMR** | 4.413 | **<0.001*** | 2.574, 7.567 |
| **EUR** | 6.371 | **<0.001*** | 3.761, 10.794 |
| **SEA** | 2.976 | **0.004*** | 1.405, 6.306 |
| **EMR** | 4.298 | **<0.001*** | 2.454, 7.526 |
| **WRP** | 3.160 | **<0.001*** | 1.780, 5.613 |
| **Urbanisation %** | | | |
| **Rural population (*reference category*)** | 1 | | |
| **Urban population** | 5.775 | **<0.001*** | 3.086, 10.808 |
| **Income Status** | | | |
| **Low (*reference category*)** | 1 | | |
| **Lower-Middle** | 1.218 | 0.535 | 0.653, 2.274 |
| **Upper-Middle** | 1.860 | **0.05*** | 1.000, 3.460 |
| **High** | 2.229 | **0.015*** | 1.165, 4.265 |

*Statistically significant [$p < 0.05$]. Regression analysis performed in STATA 17 software for statistics and data analysis.

[1] The workforce data are based on the latest available data in the NHWA data platform as of 31 March 2021, apart from the data for 2019, for which a combination of the latest available data from the NHWA data platform and WHO/King's College London survey was used.

**Table 4. Regression analysis of the combined OHWF per 10,000 population by region.**

| Density of combined Oral Heath Workforce[1,2] | Rate Ratio | Sig. | 95% conf. interval |
|---|---|---|---|
| **WHO Region** | | | |
| **AFR (*reference category*)** | 1 | | |
| **AMR** | 2.773 | **<0.001*** | 1.732, 4.441 |
| **EUR** | 3.949 | **<0.001*** | 2.511, 6.212 |
| **SEA** | 2.581 | **0.005*** | 1.327, 5.020 |
| **EMR** | 2.327 | **0.002*** | 1.378, 3.928 |
| **WRP** | 2.860 | **<0.001*** | 1.740, 4.702 |
| **Urbanisation %** | | | |
| **Rural population (*reference category*)** | 1 | | |
| **Urban population** | 6.570 | **<0.001*** | 3.368, 12.819 |
| **Income Status** | | | |
| **Low (*reference category*)** | 1 | | |
| **Lower-Middle** | 1.452 | 0.226 | 0.793, 2.658 |
| **Upper-Middle** | 2.332 | **0.007*** | 1.264, 4.304 |
| **High** | 3.762 | **<0.001*** | 1.979, 7.152 |

*Statistically significant [$p < 0.05$]. Regression analysis performed in STATA 17 software for statistics and data analysis.

[1] The workforce data are based on the latest available data in the NHWA data platform as of 31 March 2021, apart from the data for 2019, for which a combination of the latest available data from the NHWA data platform and WHO/King's College London survey was used.

[2] Oral Health Workforce: dentists, dental assistants/therapists and dental prosthetic technicians.

**Table 5. Perceived challenges globally by country income status [n = 95].**

| Item | Income status[1] | | | | | |
|---|---|---|---|---|---|---|
| | LOW | LOWER MIDDLE | UPPER MIDDLE | HIGH | ANOVA test | OVERALL SCORE |
| Maldistribution of the workforce [ex. Urban, Rural] | 8.74 | 8.06 | 6.83 | 6.18 | **0.013*** | **7.39** |
| Oral health is considered low priority | 8.65 | 7.91 | 6 | 5.93 | **0.028*** | **7.04** |
| Lack of financial support for oral health workforce training institutions | 7.95 | 7.47 | 6.26 | 4.57 | **0.001*** | **6.49** |
| Lack of continuing professional development opportunities | 7.74 | 6.28 | 4.26 | 4.03 | **0.001*** | **5.47** |
| Existence of unregistered providers of dentistry, e.g. quacks and traditional healers | 7 | 5.2 | 3.41 | 2.67 | **<0.001*** | **4.45** |
| Lack of diversity [skill mix] in the dental team | 6.58 | 6.13 | 5.43 | 4.93 | 0.175 | **5.73** |
| Poor quality dental care | 6.47 | 4.65 | 3.73 | 3.04 | **<0.001*** | **4.34** |
| Lack of workforce data for planning | 6.37 | 6.71 | 6 | 4.75 | 0.175 | **5.94** |
| Training too <u>many</u> dentists | 5.58 | 3.72 | 5.91 | 3.9 | **0.008*** | **4.59** |
| Limited jobs [vacancies] for dentists | 5.53 | 6.52 | 5.22 | 4.14 | 0.150 | **5.37** |
| Training too <u>few</u> dental assistants and therapists | 5.42 | 5.87 | 6.64 | 5.46 | 0.436 | **5.85** |
| Training too <u>few</u> dentists | 5.05 | 5.87 | 3.91 | 4.38 | 0.129 | **4.85** |
| Limited jobs [vacancies] for dental assistants and therapists | 4.42 | 6.2 | 5.27 | 3.96 | 0.084 | **5.03** |
| Migration of dentists <u>out</u> of the country | 4.26 | 4.1 | 3.17 | 3.25 | 0.355 | **3.68** |
| Training too <u>many</u> dental assistants and therapists | 4.21 | 3.84 | 2.89 | 3.26 | 0.443 | **3.56** |
| Migration of dentists <u>into</u> the country | 3.95 | 2.86 | 4.1 | 4.55 | 0.088 | **3.84** |

*Statistically significant [p<0.05]. ANOVA test was performed using Statistical Package for the Social Sciences [SPSS] software version 27.

[1.] Agreement that this is a challenge rated on a scale of 01–10 with 10 being the highest level of agreement

**Source:** WHO/King's College London survey 2019 data.

### Possible solutions

The three most highly supported potential solutions were to 'strengthen oral health policy', create 'workforce incentives to work in undeserved areas', and 'improve health workforce data for planning' [Table 6]. Countries across the income gradient were generally very strongly supportive of 'strengthening oral health policy'. Thirteen items had strong or very strong support from low-income countries, 11 of which showed a significant trend by income status: 'financial support for dental personnel education', 'regulating unregistered providers of dentistry', 'regulation of dental education' and 'creating jobs for dentists'. Interestingly, with higher income status, potential solutions were less governance focused [regulation of unregistered providers of dentistry or dental education] or related to job creation and mobility.

Amongst the textual open responses, the need for 'effective governance', including better 'regulation of education' and 'increased skill mix utilisation' were highlighted as important workforce solutions, together with the need for 'integration' of oral health with general health, and the importance of 'reorientation to prevention'.

### Discussion

This paper maps the OHWF comprehensively in support of the global strategy on oral health [1]. It highlights substantial disparities in distribution, growth, and composition, with evidence of social, regional, and urban/rural inequity. Higher-income countries appear to be making greater use of workforce skill mix than low- and lower-middle-income countries [33], based on available data. Country income status and population urbanization are strong predictors of

**Table 6. Potential solutions globally by country income status [n = 95].**

| Item | Income status[1] | | | | | |
| --- | --- | --- | --- | --- | --- | --- |
| | LOW | LOWER MIDDLE | UPPER MIDDLE | HIGH | ANOVA test | OVERALL SCORE |
| **Strengthen oral health policy** | 9.67 | 8.87 | 8.52 | 7.71 | 0.085 | **8.61** |
| **Improve health workforce data for planning** | 9.58 | 8.22 | 8.04 | 6.82 | **0.030*** | **8.05** |
| **Workforce incentives to work in underserved areas** | 9.47 | 8.41 | 8.29 | 7.04 | **0.013*** | **8.22** |
| **Financial support for dental personnel education** | 9.42 | 8.31 | 8 | 6 | **<0.001*** | **7.85** |
| **Strengthened quality of dental care** | 9.37 | 7.87 | 7.5 | 6.53 | **0.020*** | **7.81** |
| **Regulation of national systems for continuing professional development** | 8.79 | 8.38 | 7.67 | 5.86 | **0.008*** | **7.6** |
| **Creating diversity [skill mix] in the oral health workforce** | 8.53 | 7.75 | 7.25 | 6.11 | **0.025*** | **7.33** |
| **Regulating unregistered providers of dentistry, e.g. quacks and traditional healers** | 8.53 | 7.65 | 6.7 | 4.88 | **0.004*** | **6.87** |
| **Regulation of dental education** | 8.53 | 8.41 | 7.48 | 4.82 | **<0.001*** | **7.24** |
| **Creating jobs for dentists** | 8.42 | 7.59 | 6.43 | 4.56 | **<0.001*** | **6.67** |
| **Training more dentists** | 7.89 | 6.19 | 4.82 | 4.71 | **0.031*** | **5.8** |
| **Creating jobs for dental assistants and therapists** | 7.56 | 7.27 | 7.64 | 5.29 | **0.009*** | **6.84** |
| **Training more dental assistants and therapists** | 7.16 | 5.81 | 7 | 6.7 | 0.480 | **6.58** |
| **Reducing migration of dentists out of the country** | 6.47 | 5.57 | 3.76 | 4.77 | 0.210 | **5.14** |
| **Training fewer dental assistants and therapists** | 3.95 | 3.92 | 3.25 | 2.76 | 0.663 | **3.46** |
| **Reducing migration of dentists into the country** | 3.68 | 3.45 | 4.78 | 4.36 | 0.284 | **4.04** |
| **Training fewer dentists** | 2.68 | 4.13 | 4.71 | 3.19 | 0.291 | **3.72** |

*Statistically significant [p<0.05]. ANOVA test was performed using Statistical Package for the Social Sciences [SPSS] software version 27.

[1.] Agreement that this is a challenge rated on a scale of 01–10 with 10 being the highest level of agreement

**Source:** WHO/King's College London survey 2019 data.

the workforce density of dentists and, more so, for the combined OHWF. Urbanization can assist to reduce population income inequality [34]. Thus, it is not surprising that retaining a qualified health workforce in rural and underserved areas continues to be a challenge for health systems worldwide [11, 35], as well as for oral health systems [16, 36], regardless of region and country income status [9, 37–40]. For lower income countries, and the region of AFR which has the majority lower-income countries, it represents a particular challenge [41]. Inequitable distribution in relation to need [17, 42, 43], may be explained by a wide range of individual, organisational, political, economic and social contextually-specific factors [43]. This includes the recognition that the free-market business model of oral health care provision is predicated on demand for, and supply of, services [42], rather than workforce regulation and planning based on a fair distribution of the workforce according to the needs of the population [44], and guiding principles for universal health coverage [8]. The prospect of financial prosperity, better infrastructure and amenities, together with individuals' personal affection for a specific area [37, 45] may further enhance the attractiveness of urban areas in line with professional expectations. As other auxiliary roles have been developed to support the role of a dentist and are often limited by health and professional policies in their ability to practice independently, it could explain their concentration in urban areas, in addition to the range of factors influencing dentists. This is an important area for further research and consideration in seeking to diversify the workforce and facilitate other members of the OHWF, and supportive cadres of health workers, to work in rural areas.

Thus, these findings suggest that to achieve UHC, innovation on tackling the OHWF distribution to support rural areas is crucial [17, 45] with the potential for models of care beyond

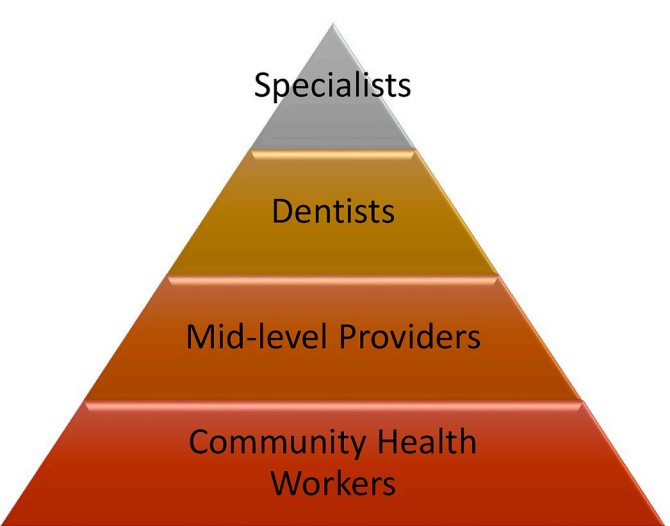

**Fig 3. Pyramid of oral healthcare delivery.** Source: Adapted from the WHO Service Organization Pyramid for an Optimal Mix of Services for Mental Health. Community Health Workers–health professionals that provide health education, referral and follow-up, case management, basic preventive health care and home visiting services to specific communities. They provide support and assistance to individuals and families in navigating the health and social services system. They do not necessarily have oral education has their primary training. Mid-level Providers–mid-level oral health workers are those who have received shorter training than dentists [between 2–4 years] but will perform some of the same tasks as dentists, such as: dental assistants, dental nurses, dental prosthetists, dental therapists and dental hygienists. Dentists: they diagnose, treat and prevent diseases, injuries and abnormalities of the teeth, mouth, jaws and associated tissues by applying the principles and procedures of modern dentistry. They use a broad range of specialized diagnostic, surgical and other techniques to promote and restore oral health. Specialists: dentists who perform post-graduate training in a specific oral health field.

the traditional focus of dentists with greater use of skill mix [17, 23, 24, 46]. Examples include the use of dental therapists and dental health aids in Alaska [47] and Tanzania [48], and using community health workers in AFR [49]. In addition, the WHO Regional Office for Africa has been supporting the development of an e-Learning course to facilitate task-sharing of oral health promotion, oral disease prevention, early detection and referral, between oral health professionals and community health workers in order to respond to the population needs [50, 51]. Innovation in healthcare should go further in line with the health pyramid framework for public health action [52] [Fig 3].

Health systems around the world are reflecting on how to build back better in light of Covid-19 [13], growing ageing populations [53], and increasing NCDs [23]. Tackling the inequitable distribution of the OHWF must be driven by a clear rationale to address the burden of oral diseases, unmet needs, and challenges to achieve UHC. In doing so, there is a great opportunity to use a range of models of care, rather than merely relying on educating and training more dentists [36]. Health systems strengthening [23], making greater use of oral health skill mix, has the potential to support UHC in delivering essential oral health services, from health promotion to prevention, treatment and rehabilitation across the life course [53], particularly in rural/remote areas [9, 41, 47]. Furthermore, having all members of primary care teams working to their full scope of practice will require planning and innovation [44], including appropriate governance.

The 2022 global strategy on oral health highlights the need for "innovative workforce models to respond to population needs", and the importance of oral health being integrated into primary healthcare [PHC] [23], providing an important opportunity for change. Creating an

NCD-ready workforce [54], will involve drawing on innovative training, utilising tools such as the global competency and outcomes framework in a context-specific manner [19], to identify, educate and train the most effective mix of staff, including community health workers, within available resources [17].

An important achievement of this research was the collation and analysis of contemporary data. Whilst data sharing is facilitated via the WHO [18], there is much to be done in relation to the OHWF.

First, we need to build comprehensive data on the complete OHWF, including non-registered personnel such as dental aides and this need for data was recognised by member states as important for the future. If they follow the pattern of higher-income countries [55], dental aides possibly exceed the volume of dentists globally and are a vitally important resource in support of quality dental care; and, we need to link these data with the wider community and primary care workforce.

Second, data should be regularly updated, with gaps and abrupt changes explained by member states in submission to the NHWF Accounts as this will go some way to building robust intelligence on the OHWF. This paper highlights the dramatic drop in dentists reported by Brazil to the NHWFA indicating significant under-reporting. Workforce experts suggest that the dentist workforce in Brazil, the highest in the world, is spread almost evenly between public and private sectors [56]. When the density suddenly fell from 12 to 6 per 10,000 in 2018 in the NHWFA [Fig 2], it suggests that only public sector data were reported. Sudden changes in trend patterns can also be credited to data updates and improved data quality [15, 18]. However, breaks in series due to metadata differences may adversely affect the global and regional results, particularly when populous countries with a large workforce are involved.

Third, there is a need for contemporary evidence on dental education providers across the public and private sectors, as the private sector appears to play an increasing role in education, particularly in lower middle-income countries. These data are not routinely collected; thus, particularly at country-level there will need to be consideration of how the evidence on education of relevant personnel is integrated.

Fourth, clarity over titles and the scope of practice for country-specific personnel being educated and trained is required. The categorisation of oral health professionals with different scopes of practice under the same group of 'dental assistants and therapists' is not ideal. Strengthening health workforce information is essential for evidence-based health workforce planning and policy [11, 16, 17], including more granular data relating to their title and specific 'scope of practice'. Global consideration of oral health workforce categories, and their local relevance within the general healthcare system, is required to further enhance the survey beyond the global health workforce accounts and refine data collection instruments to provide comprehensive and comparable data that can be used at national, regional and global levels.

Fifth, data on workforce demography [age, gender, etc], sector of employment, inflow and outflow, and their level of contribution to public and private sectors should ideally also be comprehensive to facilitate oral health planning [16], particularly as the private sector plays a major role in delivering oral and dental care in many countries [57].

Sixth, and finally, this timely paper supports the need for reform, with the global strategy calling for the development of 'innovative workforce models', together with an expansion of 'competency-based education' in order to address population needs [23, 58]. The challenges and possible solutions are clearly outlined, with WHO-led policies and goals now in place to support local policy and action [59]. Efforts should focus on the capability of different OHWF categories; and their scope of practice particularly to deliver UHC to underserved rural/remote populations and in doing so address governance challenges highlighted in this paper. This should include being part of the entire PHC team to provide oral health care but also be

engaged in overall management of patients at health facility level, while reassessing and updating national regulatory policies within the broader national health workforce strategy [23, 59]. Additionally, countries with high density of dentists should look at their dental education systems as a matter of urgency [9, 56, 60], and consider how the workforce might helpfully be diversified, as well as how oral health training should be more systematically integrated with general health education systems. Resource and workforce planning models need to inform better alignment of workforce education and training, with public health goals and population oral health needs, together with the importance of ensuring that jobs exist within health systems for those being educated and trained. This will also require effective governance, planning and resourcing. The publication of country profiles as part of the Global Oral Health Status Report provides crucial information to support all nations in preparing for active response [8].

## Conclusion

The distribution of the global OHWF is inequitable between and within countries, with variable, and limited, use of skill mix. Challenges and solutions vary by country income status. Innovative models are required to address oral health needs and then to support achievement of UHC for all, supported by robust data. These should involve utilisation of workforce skill mix, task-sharing, and leverage primary care workers including community health workers to strengthen the oral health agenda globally.

## Supporting information

**S1 Text. Definition of oral health workforce categories.**
(DOCX)

**S2 Text. The WHO global oral health workforce survey questionnaire.**
(DOCX)

**S1 Table. Global data availability per country and professional category.**
(DOCX)

**S2 Table. Global dataset by country: density of dentists per 10,000 population globally, 2000–2019.**
(DOCX)

**S3 Table. Dental schools [public and private] by region and country income status.** Note: further information available on request from authors.
(DOCX)

**S1 Appendix. Trends in density of dentists per 10,000 population globally, 2000–2019.**
(DOCX)

## Acknowledgments

All authors would like to acknowledge the National Health Workforce Accounts [NHWA] initiative, thank the colleagues in the WHO Health workforce department and representatives of member states who contributed to the global survey. And others, who commented on the protocol and questionnaire, particularly Teena Kunjumen from the WHO.

**Disclaimer:** The authors are responsible for the views expressed in this article and they do not necessarily represent the views, decisions, or policies of the institutions with which they are affiliated".

## Author Contributions

**Conceptualization:** Jennifer E. Gallagher, Grazielle C. Mattos Savage, Yuka Makino, Benoit Varenne.

**Data curation:** Jennifer E. Gallagher, Grazielle C. Mattos Savage.

**Formal analysis:** Grazielle C. Mattos Savage, Sarah C. Crummey, Wael Sabbah.

**Investigation:** Jennifer E. Gallagher, Grazielle C. Mattos Savage.

**Methodology:** Jennifer E. Gallagher, Grazielle C. Mattos Savage, Sarah C. Crummey, Wael Sabbah, Yuka Makino, Benoit Varenne.

**Project administration:** Jennifer E. Gallagher, Sarah C. Crummey, Yuka Makino.

**Resources:** Jennifer E. Gallagher.

**Supervision:** Jennifer E. Gallagher, Wael Sabbah.

**Validation:** Yuka Makino.

**Visualization:** Grazielle C. Mattos Savage.

**Writing – original draft:** Jennifer E. Gallagher, Grazielle C. Mattos Savage, Sarah C. Crummey.

**Writing – review & editing:** Jennifer E. Gallagher, Grazielle C. Mattos Savage, Sarah C. Crummey, Wael Sabbah, Yuka Makino, Benoit Varenne.

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
