## [Decision Letter · Decision Letter 0]

28 Feb 2023

PONE-D-22-35221Oral Health Workforce Inequity: opportunity for actionPLOS ONE

Dear Dr. Gallagher

Thank you for submitting your manuscript to PLOS ONE. After careful consideration, we feel that it has merit but does not fully meet PLOS ONE’s publication criteria as it currently stands. Therefore, we invite you to submit a revised version of the manuscript that addresses the points raised during the review process.

Please submit your revised manuscript by Apr 14 2023 11:59PM. If you will need more time than this to complete your revisions, please reply to this message or contact the journal office at plosone@plos.org. Please include the following items when submitting your revised manuscript:A rebuttal letter that responds to each point raised by the academic editor and reviewer(s). You should upload this letter as a separate file labeled 'Response to Reviewers'.A marked-up copy of your manuscript that highlights changes made to the original version. You should upload this as a separate file labeled 'Revised Manuscript with Track Changes'.An unmarked version of your revised paper without tracked changes. You should upload this as a separate file labeled 'Manuscript'.

We look forward to receiving your revised manuscript.

Kind regards,

Azmeraw Ambachew Kebede, MSc

Academic Editor

PLOS ONE

Journal Requirements:

6. Please upload a copy of Supporting Information Figures 1Sa,b and Tables 1Sa,b which you refer to in your text on page 14 and 11.

Reviewers' comments:

Reviewer's Responses to Questions

**Comments to the Author**

1. Is the manuscript technically sound, and do the data support the conclusions?

Reviewer #1: Yes

2. Has the statistical analysis been performed appropriately and rigorously? 

Reviewer #1: Yes

3. Have the authors made all data underlying the findings in their manuscript fully available?

Reviewer #1: Yes

4. Is the manuscript presented in an intelligible fashion and written in standard English?

Reviewer #1: Yes

5. Review Comments to the Author

Reviewer #1: PONE-D-22-35221

Oral Health Workforce Inequity: opportunity for action

PLOS ONE

Dear Authors,

Your manuscript is really interesting and well conducted, and I only have 2 comments in the Methods (see below). The rest of the paper looks fine to me.

METHODS

Page 7, lines 178-181: I suggest that you add a description in this paragraph of what type of descriptive analysis you performed to report on OHWF density. Measures of frequency, central tendency, variation, position?

Page 7, lines 181-182: Please provide more details on the thematic analysis approach that you took. There are multiple unique guides in the literature for performing a thematic analysis, which one(s) did you uses? Please cite this source. Reading your results, I could not tell if you performed a thematic analysis or a qualitative content analysis. Please review the steps you took to perform the qualitative analysis, and add 1-2 sentences detailing your methods for this analysis.

Thank You,

Kind Regards

6. PLOS authors have the option to publish the peer review history of their article (what does this mean?). If published, this will include your full peer review and any attached files.

Reviewer #1: No

---

## [Author Response · Author response to Decision Letter 0]

3 Jul 2023

Extracted from covering letter:

A. Editorial comments

• A marked-up copy of your manuscript that highlights changes made to the original version has been uploaded as a separate file labeled 'Revised Manuscript with Track Changes'.

• An unmarked version of your revised paper without tracked changes is labeled 'Manuscript'.

o Both versions uploaded, as requested.

o N/A.

o We trust that we have followed PLOS ONW style requirements in the revised version of the manuscript.

o Dataset provided as a supplementary file

o Option B has been taken and data uploaded as S4

o Thank you.

• Not required as a data set is now available as a supplementary file S4.

• Captions for figures have been inserted in the manuscript, as requested.

• Captions for supplementary files have been inserted at the end of the manuscript, as requested.

6. Please upload a copy of Supporting Information Figures 1Sa,b and Tables 1Sa,b which you refer to in your text on page 14 and 11.

• Uploaded as requested.

B. Reviewer #1: Yes

Comments to the Author

1. Is the manuscript technically sound, and do the data support the conclusions?

Reviewer #1: Yes

2. Has the statistical analysis been performed appropriately and rigorously? 

Reviewer #1: Yes

3. Have the authors made all data underlying the findings in their manuscript fully available?

• Data set available as a supplemental file S4.

4. Is the manuscript presented in an intelligible fashion and written in standard English?

Reviewer #1: Yes

5. Review Comments to the Author

Reviewer #1: PONE-D-22-35221

Oral Health Workforce Inequity: opportunity for action

PLOS ONE

Dear Authors,

Your manuscript is really interesting and well conducted, and I only have 2 comments in the Methods (see below). The rest of the paper looks fine to me.

• Thank you.

METHODS

Page 7, lines 178-181: I suggest that you add a description in this paragraph of what type of descriptive analysis you performed to report on OHWF density. Measures of frequency, central tendency, variation, position? 

• we used the data provided by OHWF accounts (Dentists (per 10 000) (who.int)) where available and replicated this where data were used from the global survey. A sentence has been added to the text to identify that the numerator is the sum of country number of dentists or wider OHWF, divided by the denominator that is the population number and reported per 10,000 population.

Page 7, lines 181-182: Please provide more details on the thematic analysis approach that you took. There are multiple unique guides in the literature for performing a thematic analysis, which one(s) did you uses? Please cite this source. Reading your results, I could not tell if you performed a thematic analysis or a qualitative content analysis. Please review the steps you took to perform the qualitative analysis, and add 1-2 sentences detailing your methods for this analysis.

• Thematic analysis of the open responses was undertaken, and brief description has been added to the Methods. 

1. Submit Figures as a high-resolution PPTX or TIF file.

• Figures are in TIF.

2. Study design and Period is not clear and it is good to say more on it.

• Further details added to the methods section.

3. ‘‘Density - Global OHWF per 10,000 population, by category and region, 2019’’ Add a picture so that the reader can get a clearer picture of the distribution.

• We can refer to the global statistics in the baseline report and global maps. 

4. Attach the questionnaire used in the study, and cite its reference, and elaborate in detail on its validity and reliability.

• The questionnaire was already attached and available in the appendix; however, this is now clearer because we have listed the supplementary files (labelled) at the end of the paper

• This questionnaire was developed in collaboration with WHO teams (oral health and human resources for oral health) against standard criteria outlined in the national workforce accounts, together with the wider literature and was tested with dental and WHO workforce experts re face validity. This is now outlined in the text. 

• Limitations are discussed. 

5. Figure 3 elaborates the challenges related to the global oral health work force over the regions. Describe these in a table with statistical analysis to highlight factor/s which significantly challenges among the lists.

• Thank you for this suggestion. We have looked at the data by region and income status and considered that the analysis by income status provided important insights in relation to each of the challenges and potential solutions and show some important trends. We have therefore replaced Figures 3 and 4 with tables which we consider enhance the content of this paper.

---

## [Decision Letter · Decision Letter 1]

25 Sep 2023

Oral Health Workforce Inequity: opportunity for action

PONE-D-22-35221R1

Dear Dr. Gallagher,

We’re pleased to inform you that your manuscript has been judged scientifically suitable for publication and will be formally accepted for publication once it meets all outstanding technical requirements.

Kind regards,

Mathew Albert Wei Ting Lim

Academic Editor

PLOS ONE

Additional Editor Comments (optional):

One of the secondary reviewers has added some additional comments that you may wish to review despite the recommendation to accept your manuscript for publication.

Reviewers' comments:

Reviewer's Responses to Questions

**Comments to the Author**

1. If the authors have adequately addressed your comments raised in a previous round of review and you feel that this manuscript is now acceptable for publication, you may indicate that here to bypass the “Comments to the Author” section, enter your conflict of interest statement in the “Confidential to Editor” section, and submit your "Accept" recommendation.

Reviewer #1: All comments have been addressed

Reviewer #2: (No Response)

2. Is the manuscript technically sound, and do the data support the conclusions?

Reviewer #1: Yes

Reviewer #2: Yes

3. Has the statistical analysis been performed appropriately and rigorously? 

Reviewer #1: Yes

Reviewer #2: Yes

4. Have the authors made all data underlying the findings in their manuscript fully available?

Reviewer #1: Yes

Reviewer #2: Yes

5. Is the manuscript presented in an intelligible fashion and written in standard English?

Reviewer #1: Yes

Reviewer #2: No

6. Review Comments to the Author

Reviewer #1: I read their revised manuscript (PONE- D-22-35221R1) I have accepted and recommend the manuscript entitled "Oral Health Workforce Inequity: opportunity for action." for publication.

With kind regards

Dr. Wondwossen Fantaye Abawello

Reviewer #2: Dear authors,

Some of the sentences were too long, and require rephrasing. I found the paper of interest. Additional comments are added in the paper.

Kind regards,

Peer-reviewer

7. PLOS authors have the option to publish the peer review history of their article (what does this mean?). If published, this will include your full peer review and any attached files.

Reviewer #1: **Yes: **Dr. Wondwossen Fantaye Abawello

Reviewer #2: No

---

## [Editor Report · Acceptance letter]

12 Mar 2024

PONE-D-22-35221R1 

PLOS ONE

Dear Dr. Gallagher, 

I'm pleased to inform you that your manuscript has been deemed suitable for publication in PLOS ONE. Congratulations! Your manuscript is now being handed over to our production team.

Kind regards, 

on behalf of

Dr Mathew Albert Wei Ting Lim 

Academic Editor

PLOS ONE